



# National-Scale Sub-meter Mapping of *Spartina alterniflora* in Mainland China 2020

Bingfeng Zhou[1,※], Meng Xu[1,※], Jinyan Tian[1,2,3,*], Mingming Jia[4], Dehua Mao[4], Kai Cheng[5], Xiumin Zhu[1], Haoyue Jiang[1], Jie Song[1], Yinghai Ke[1], Zhenxin Zhang[2], Yue Huang[1], Miaojing Wei[6], Lin Zhu[1], Xiaojuan Li[1,2], Huili Gong[1,2]

[1]Beijing Laboratory of Water Resources Security, Capital Normal University, Beijing, 100048, China
[2]Key Laboratory of 3D Information Acquisition and Application, Ministry of Education, Capital Normal University, Beijing, 100048, China
[3]Jing-Jin-Ji Geospatial Data Center, Capital Normal University, Beijing, 100048, China
[4]Key Laboratory of Wetland Ecology and Environment, Northeast Institute of Geography and Agroecology, Chinese Academy of Sciences, Changchun, 130102, China
[5]Institute of Remote Sensing and Geographic Information System, School of Earth and Space Sciences, Peking University, Beijing, 100871, China
[6]Graduate School of Architecture, Planning and Preservation, Columbia University, New York, 10027, USA

※These authors contributed equally to this work.

*Correspondence to*: Jinyan Tian (tjyremote@126.com)

**Abstract.** Current large-scale maps of *Spartina alterniflora* (*S. alterniflora*) with 10 m resolution hinder accurate delineation of community boundaries, detection of internal features such as creeks, and identification of small patches. These limitations further compromise the accuracy of spatial distribution extraction and subsequent analyzes. To this end, this study produced the first 2020 national-scale Sub-meter *S. alterniflora* Map of Mainland China (CM-SSM), using an object- and sub-meter-enhanced pixel-based phenological feature composite method. The method integrates phenological features from Sentinel-2 with spatial and textural details from Google Earth imagery, improving the spectral separability and mitigating mixed-pixel effects. Compared to the 10 m *S. alterniflora* product of Mainland China (CMSA), CM-SSM improved overall accuracy by 14.60 % and the F1 score by 0.21. Although the total mapped areas of CM-SSM (59,371 ha) and CMSA (58,006 ha) differ by only 1,365 ha, their spatial distributions diverge substantially. When benchmarked against CM-SSM, CMSA exhibited commission and omission errors totaling 34,273 ha (57.73 %). Moreover, the number of patches identified by CM-SSM (148,072) was over 17 times greater than that of CMSA, reflecting its superior capability in detecting fragmented distributions. In addition, Soil Organic Carbon (SOC) estimates derived from CM-SSM were 706.69 Gg (23.09 %) higher than those reported by the corresponding national SOC product for the same year, emphasizing the essential contribution of high-resolution mapping to accurate carbon accounting for *S. alterniflora*. These advances enhance understanding of *S. alterniflora* invasion dynamics, support carbon accounting, and inform evidence-based coastal wetland management and restoration. The map is available at https://doi.org/10.5281/zenodo.16296823 (Xu et al., 2025).



# 1 Introduction

*Spartina alterniflora* (*S. alterniflora*), native to the Atlantic coast of North America and widely recognized as a classic
"ecosystem engineer", was intentionally introduced into China in 1979 to enhance embankment stability and mitigate coastal
erosion (Jackson et al., 2021; Liu et al., 2020). Due to its lack of natural predators and high reproductive capacity, *S. alterniflora* has rapidly expanded along China's coast over the past four decades, reaching a total extent of more than 50,000
ha (Meng et al., 2020). This rapid spread has posed serious threats to coastal ecosystems, including the displacement of
native species, degradation of nearshore habitats, and measurable declines in biodiversity (Li et al., 2009; Okoye et al., 2020).
In response, a range of control strategies, including physical removal, chemical control, and biological replacement, were
employed to manage the invasion of *S. alterniflora* in China (Zheng et al., 2023a). However, the effectiveness of these
measures varied significantly across regions, and the risk of reinvasion remained high (Li et al., 2022a; Zhao et al., 2020).
Given the rapid natural spread of *S. alterniflora* and the continued influence of human activities on its spatial distribution,
there is an urgent need for accurate, large-scale monitoring to delineate its distribution patterns and assess the effectiveness
of removal effort.

Remote sensing has been widely used for mapping *S. alterniflora* due to its capability for large-area coverage and long-term,
repeatable monitoring (Chen et al., 2020; Lourenço et al., 2021; Lv et al., 2019). Existing mapping products can be divided
into two categories based on spatial resolution. The first category includes products derived from High Resolution (HR)
imagery (10–30 m), such as Landsat and Sentinel series data (Zuo et al., 2025; Zuo et al., 2012). Liu et al. (2018) produced
the *S. alterniflora* map along mainland China's coast in 2015 at 30 m using Landsat 8. Subsequently, Hu et al. (2021)
generated a 2019 coastal saltmarsh map (including *S. alterniflora*) using Sentinel-1, with an improved spatial resolution from
30 m to 10 m. To analyze expansion dynamics of *S. alterniflora*, Zhang et al. (2017) mapped a sparse multi-temporal dataset
spanning 1990 to 2014 using Landsat time-series imagery. Similarly, Mao et al. (2019) produced *S. alterniflora* maps at 5- or
10-year intervals from 1990 to 2015. The mapping intervals in previous studies were typically greater than 5 years, making it
difficult to capture the ongoing spread of *S. alterniflora*. Therefore, Li et al. (2024) generated annual distribution maps of *S. alterniflora* between 2017 and 2021 using Sentinel-2 imagery, enabling precise monitoring of interannual changes.

Nevertheless, relying solely on HR imagery (10-30 m) presents three key challenges due to inherent spatial limitations. First,
existing products often fail to detect small patches of *S. alterniflora*, which are ecologically important and may function as
early indicators of invasion risk (Chen et al., 2020). Second, boundaries between *S. alterniflora* and co-occurring species
remain poorly defined, limiting accurate delineation of its spatial extent. For instance, in mangrove areas, boundary
misclassification hinders reliable assessment of invasion risk (Zheng et al., 2023b). Third, internal features within *S. alterniflora* communities, such as creeks and open spaces, are difficult to capture. Creeks, in particular, influence
hydrological processes and seed dispersal, thereby affecting the rate and extent of spread (Sun et al., 2020). These issues
obstruct the effective application of existing products in precise monitoring and ecological management.





The second category includes products derived from Very High Resolution (VHR) imagery (finer than 10 m), such as UAV (Windle et al., 2023), WorldView (Dong et al., 2024), Gaofen (Li et al., 2021), and SPOT imagery (Liu et al., 2017a). Compared with HR data, these sources offer improved spatial detail and effectively reduce classification errors caused by mixed pixels. However, their high acquisition cost and limited spatial coverage constrain their use in large-scale applications. The Google Earth (GE) platform provides free access to sub-meter imagery with rich spatial detail (Li et al., 2022b), creating

new opportunities for large-area, high-precision monitoring of *S. alterniflora*. Nevertheless, the lack of multispectral information in GE imagery restricts its utility for spectral-based identification (Zhou et al., 2024).

Phenology-based methods, which exploit spectral variations during vegetation growth, are widely recognized for their effectiveness in mapping *S. alterniflora* (Zeng et al., 2022; Zhang et al., 2022). Initially, single-date imagery from the growing period of *S. alterniflora* was used for classification (Ouyang et al., 2013; Wang et al., 2015). However, relying

solely on the growing period is insufficient for accurate mapping because *S. alterniflora* shares similar spectral features with other saltmarsh vegetation, especially evergreen mangroves (Ai et al., 2017; Sun et al., 2021). To address this, time-series imagery has been used to construct phenological trajectories, enabling improved distinction through the integration of multiple growth phases (Sun et al., 2016; Liu et al., 2017b). For example, Sun et al. (2016) constructed monthly NDVI time-series from the Chinese HuanJing-1 satellite imagery to monitor salt marsh vegetation, including *S. alterniflora*. While this

method shows strong potential, two major challenges remain. Frequent cloud cover and tidal disturbances in coastal regions complicate the acquisition of high-quality, high-temporal-resolution imagery. Moreover, *S. alterniflora* exhibits spatial phenological heterogeneity, where communities in different regions may be at different phenological stages simultaneously, introducing spectral inconsistencies that reduce classification stability and accuracy.

To address the limitations mentioned above, Tian et al. (2020a) proposed a Pixel-based Phenological Feature (PPF)

composite method. This method leverages the computational capacity and extensive data resources of the Google Earth Engine (GEE) platform to perform image compositing at the pixel level, aiming to overcome the difficulty of acquiring high-quality imagery in intertidal zones. Instead of relying on entirely cloud-free scenes, the method integrates all cloud-free pixels across multiple images, significantly improving the utilization of available data and alleviating the scarcity of usable imagery. Two key phenological periods (green period and senescence period) were selected based on their spectral

distinctiveness. During the green period, *S. alterniflora* is spectrally distinguishable from non-vegetated surfaces such as mudflats and water; in the senescence period, it is more separable from evergreen species like mangroves. These periods provide complementary spectral features, enhancing the separability of *S. alterniflora* and capturing most of its relevant phenological information, while minimizing interference from transitional periods. Furthermore, to account for the phenological variability of *S. alterniflora* across different geographic locations, the method constructs composite imagery by

extracting the greenest pixels during the green period and the most senescent pixels during the senescence period. This selection of extreme phenological states effectively reduces spatial heterogeneity in the phenology of *S. alterniflora*. Subsequent studies have confirmed the value of the dual-temporal phenological feature composite method. It has been shown to improve *S. alterniflora* classification accuracy (Zhang et al., 2020; Zhang et al., 2023) and to perform well in



broader coastal wetland mapping (Chen and Kirwan et al., 2022; Zhao et al., 2023), demonstrating its effectiveness and
reliability. However, these studies relied on pixel-based classification methods, which consider only the spectral value of
individual pixels and ignore spatial relationships with neighboring pixels, thus limiting classification accuracy (Frohn et al.,
2011). Independent pixel-wise classification often results in isolated misclassified pixels, producing the salt-and-pepper
effect (Dronova, 2015). In contrast, object-based image analysis (OBIA) integrates shape, texture, and spatial context
features and takes advantage of the spectral consistency within image objects. This method has shown promising potential
for *S. alterniflora* identification (Wang et al., 2021). Nevertheless, the application of OBIA in large-scale, sub-meter *S. alterniflora* mapping using VHR imagery remains underexplored.

This study aims to establish an object-based, large-scale mapping approach by integrating multi-source remote sensing
imagery to produce the first sub-meter map of *S. alterniflora* across mainland China. Our objectives are threefold: (1) to
develop a novel Object- and Sub-meter-enhanced PPF (OSPPF), (2) to produce a 2020 Sub-meter *S. alterniflora* Map of
Mainland China (CM-SSM), and (3) to evaluate the significant improvements of CM-SSM over the latest 10 m resolution
map of *S. alterniflora*. Overall, the OSPPF effectively mitigates the mixed-pixel problem by incorporating sub-meter GE
imagery. Additionally, by integrating OBIA, the proposed OSPPF approach leverages multi-dimensional features such as
texture, shape, and spatial context, thereby overcoming the limitations of pixel-based classification that relies solely on
spectral features. This integration enhances the classification accuracy of *S. alterniflora*. CM-SSM addresses issues in
existing products, such as inaccurate boundary depiction of *S. alterniflora*, poor identification of internal details, and limited
ability to detect small patches. As a high-resolution and high-accuracy map, CM-SSM provides a robust data foundation for
management assessment, blue carbon stock estimation, and coastal sustainable development.

## 2 Materials and methods

### 2.1 Study area

*S. alterniflora* is distributed across nine provinces in mainland China (20° N–41° N, 108° E–122° E), with coverage areas
ranked in descending order as follows: Shanghai, Jiangsu, Zhejiang, Fujian, Shandong, Guangxi, Guangdong, Tianjin, and
Hebei. Common co-occurring species include *Phragmites australis*, *Suaeda salsa*, *Tamarix chinensis*, and mangroves.

To preliminarily delineate the potential distribution of *S. alterniflora*, a 10 km coastal buffer zone was generated by
extending seaward from the coastline dataset. Considering the regional differences in co-occurring species and phenology,
the coastal buffer was subdivided into five subregions: the Southern Coastal Zone (SCZ), Yangtze River Estuary Coastal
Zone (YRECZ), Jiangsu Coastal Zone (JSCZ), Yellow River Delta Coastal Zone (YRDCZ), and Northern Coastal Zone
(NCZ). Subsequently, two national-scale *S. alterniflora* products from 2020 (see Sect. 2.2.2) were collected to extract their
union, which was then expanded with a 100 m buffer to cover potential edge areas. Omission errors were manually corrected
through visual interpretation of VHR imagery. The red-highlighted area represented an *S. alterniflora* patch. The study area





was divided into five subregions, each containing multiple *S. alterniflora* patches. All Sentinel-2 and GE imagery were collected within these subregions (Fig. 1).

**Figure 1: Location of the study site in the coastal zone of mainland China. The background imagery is provided by Esri (https://www.esri.com) and its data partners. The VHR imagery in the figure is from © Google Earth 2020.**



## 2.2 Datasets

### 2.2.1 Remote sensing imagery

This study acquired approximately 6,007 Sentinel-2 Surface Reflectance (SR) images from the year 2020 through the GEE platform. As *S. alterniflora* grows in intertidal zones and is highly susceptible to cloud contamination and tidal variation, both scene-based and pixel-based methods were applied to ensure image quality (Chen et al., 2025). At the scene level, images with more than 70% cloud cover within the study area were excluded based on metadata attributes (Ni et al., 2021). At the pixel level, bitwise operations were used to the Sentinel-2 Scene Classification Layer (SCL) to mask cloud (SCL = 7–9), cirrus (SCL = 10), and cloud shadows (SCL = 3). To further reduce tidal effects on classification, water pixels (SCL = 6) were also removed. Additionally, 0.9 m GE imagery from 2020 with RGB bands was selected under low-tide and cloud-free conditions. In regions lacking high-quantity 2020 imagery, supplementary GE imagery from 2019 or 2021 was used.

### 2.2.2 Existing *S. alterniflora* products

Two *S. alterniflora* products covering coastal mainland China in 2020 were collected for study area delineation and comparative analysis (Table 1). Mao et al. (2019) developed a multi-temporal *S. alterniflora* dataset (1990–2015) and used it to generate a 30 m resolution product in 2020 (hereafter referred to as SpProduct_30m). More recently, Li et al. (2024) generated a 10 m resolution *S. alterniflora* map of mainland China for 2020 (CMSA) using Sentinel-2 imagery. Both SpProduct_30m and CMSA served as critical references for delineating the study area (see Sect. 2.1). As the highest-resolution national-scale *S. alterniflora* product for 2020, CMSA was used as the benchmark. Specifically, we evaluated the improvement in detection capability achieved by our sub-meter product relative to the CMSA.

**Table 1 Details of existing large-scale *S. alterniflora* products in 2020.**

| Product | Dataset | Resolution | Extent | References |
|---|---|---|---|---|
| SpProduct_30m | Landsat-8 | 30 m | Mainland China | Mao et al. (2019) |
| CMSA | Sentinel-2 | 10 m | Mainland China | Li et al. (2024) |

### 2.2.3 Reference data

The ecological complexity of intertidal zones where *S. alterniflora* grows poses challenges for large-scale field sampling and limits the availability of sufficient reference data. To address this issue, we constructed a high-quality sample dataset by integrating field surveys with multi-source VHR imagery. First, field surveys were conducted in 2020 in typical *S. alterniflora* habitats, including the Beibu Gulf, Jiulong River Estuary, and Zhangjiang Estuary. Using differential GPS, we collected 1,396 ground validation points, including 661 *S. alterniflora* and 735 non-*S. alterniflora* samples. Subsequently,



VHR imagery temporally aligned with the field survey was collected from UAVs, the Gaofen series, and GE. Based on this, experienced researchers visually interpreted and manually labeled 10,570 *S. alterniflora* and 20,247 non-*S. alterniflora* points by overlaying field data with VHR imagery. The non-*S. alterniflora* class includes co-occurring species (e.g., mangroves, *Phragmites australis*, *Suaeda salsa*) and other land covers such as mudflats. All sample points were evenly

distributed across the five subregions defined in Sect. 2.1. Finally, a stratified random sampling method was used to divide *S. alterniflora* and non-*S. alterniflora* sample points within each subregion into training and validation sets at a 7:3 ratio. In addition, the class ratio of *S. alterniflora* to non-*S. alterniflora* (approximately 1:2) was preserved during the splitting process. The sample points from all subregions were then merged to form a complete reference dataset. This method mitigated issues of spatial autocorrelation and class imbalance (Wang et al., 2020). The training set was used to select input

features for the Random Forest (RF) classifier (see Sect. 2.3.3), while the validation set was employed to assess classification accuracy.

## 2.3 Development of an Object- and Sub-meter-enhanced PPF

This study proposed an Object- and Sub-meter-enhanced Pixel-based Phenological Feature (OSPPF) composite method for mapping *S. alterniflora*, including four steps (Fig. 2). First, a Pixel-based Phenological Feature (PPF) was constructed using

Sentinel-2 imagery. Second, spatial and textural features extracted from GE imagery were integrated to enhance the PPF, resulting in the Sub-meter-enhanced PPF (SPPF). Third, a multi-scale object-based segmentation strategy was used to extract the OSPPF. Finally, a RF classifier was applied to generate the initial result, which was then manually refined to generate the final *S. alterniflora* distribution map.



**(a) Pixel-based phenological festure composite method**

Sentinel-2 SR data during 2020

↓

Cloud and water masking

↓

Conducting phenological analysis of *S. alterniflora* by region

↓

Senescence period    Green period

↓

Select original spectral bands and index-derived bands

↓

Pixel-based phenological features composite image (15 bands)

**(b) Integration of fine-scale spatial information in PPF**

GE imagery

↓

Extract spectral features    Extract texture features    RGB bands

↓

Spatial registration up-sampling → Image compositing in GEE

↓

The image incorporating sub-meter phenological spectral features (24 bands, 0.9 m)

**(d) Object-based classification**

Reference data

↓

Object-based random forest classifier

↓

Distribution of *S. alterniflora*

**(c) Integration of object-based spatial features in SPPF**

Multi-scale optimization segmentation

↓

Object-based and Sub-meter-enhanced Pixel-based Phenological Feature

**Figure 2: Workflow of the proposed OSPPF.**

**2.3.1 Pixel-based phenological feature composite method**

Phenological features are critical for identifying *S. alterniflora*. Previous studies have shown that the green and senescence periods are two key phenological phases that enhance the spectral separability of *S. alterniflora* from background land covers (Tian et al., 2020). Accordingly, this study constructed annual NDVI time series curves based on Sentinel-2 imagery acquired between January 1 and December 31, 2020, to determine the green and senescence periods for each subregion. Using JSCZ as an example, 175 pure *S. alterniflora* pixels were selected through visual interpretation of GE imagery, ensuring an even spatial distribution. NDVI values for these pixels were calculated from cloud-masked Sentinel-2 imagery, extracting the median value for each Day of Year (DoY) to construct the NDVI time series. To reduce noise caused by cloud cover and atmospheric effects, the NDVI time series was smoothed using the Savitzky-Golay (SG) filter (Savitzky and Golay, 1964). As shown in Fig. 3, NDVI values below 0.3 during DoY 1–125 indicated the senescence period, whereas values above 0.5 during DoY 200–310 corresponded to the green period. Following this procedure, the key phenological windows were identified for each subregion (Table 2).



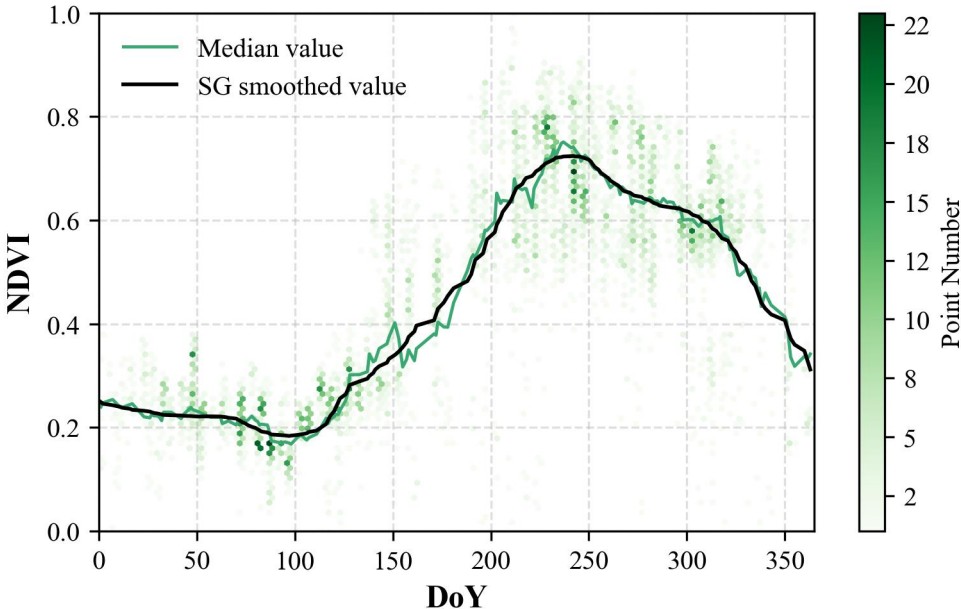

**Figure 3: NDVI time series analysis of the *S. alterniflora* in JSCZ. Point density is represented using hexagonal binning, with color intensity indicating the concentration of data points.**

**Table 2 Sub-regional phenological windows.**

| Sub-region | Dominant companion species | Senescence period (DoY) | Green period (DoY) |
|---|---|---|---|
| NCZ | *T. chinensis* | 1–150 | 200–290 |
| YRDCZ | *P. australis, S. salsa, T. chinensis* | 1–175 | 200–325 |
| JSCZ | *S. salsa, P. australis* | 1–125 | 200–310 |
| YRECZ | *P. australis, S. mariqueter* | 1–105 | 205–315 |
| SCZ | Mangroves | 1–140 | 160–300 |

Based on the two identified phenological periods, the PPF was constructed by integrating vegetation indices and original spectral bands. Specifically, five indices were selected to characterize the phenological periods (Table 3): Normalized Difference Vegetation Index (NDVI), Enhanced Vegetation Index (EVI), Plant Senescence Reflectance Index (PSRI), Normalized Difference Water Index (NDWI), and Land Surface Water Index (LSWI). NDVI, EVI, and NDWI were used during the green period, while PSRI and LSWI characterized the senescence period. The selection rationale is as follows: (1) NDVI is sensitive to green vegetation but tends to saturate under dense canopies, whereas EVI remains responsive at high biomass levels, making them complementary during the green period (Ni et al., 2021). (2) NDWI captures reflectance differences between vegetation and water in the green and near-infrared bands, effectively distinguishing *S. alterniflora* from water when canopy cover is high during the green period (Mancino et al., 2020). (3) PSRI responds to changes in carotenoid



pigments associated with senescence, making it suitable for detecting vegetation during the senescence period (Tian et al., 2020). (4) As LSWI is sensitive to leaf water content, the progressive moisture loss in *S. alterniflora* during senescence leads to decreased LSWI values, improving its separability from moist backgrounds such as mudflats and water (Wu et al., 2020).

Each index was derived from Sentinel-2 imagery of its corresponding phenological period, with median values of valid observations computed per pixel to generate the composite images.

In addition, five original bands of Sentinel-2 were selected for both phenological periods: B2 (blue), B3 (green), B4 (red), B8 (NIR) and B11 (SWIR 1). B2, B3, and B4 cover the visible spectrum and are useful for distinguishing *S. alterniflora* from water and mudflats. B8 and B11 are sensitive to vegetation structure and moisture content, effectively capturing

spectral transitions of *S. alterniflora* from the green to senescence period. Finally, the vegetation indices and selected spectral bands for both phenological periods were integrated to construct the PPF composite images.

**Table 3 Vegetation indices used in this study.**

| Vegetation index | Formula | Reference |
|---|---|---|
| NDVI | (NIR - Red) / (NIR + Red) | Rouse et al. (1974) |
| EVI | $2.5 \times$ (NIR - Red)/(NIR + 6 $\times$ Red - 7.5 $\times$ Blue + 1) | Huete et al. (2002) |
| PSRI | (Red - Blue) / NIR | Merzlyak et al. (1999) |
| NDWI | (Green - NIR) / (Green + NIR) | Gao et al. (1996) |
| LSWI | (NIR - SWIR1) / (NIR + SWIR1) | Chandrasekar et al. (2010) |

### 2.3.2 Integration of fine-scale spatial information in PPF

To address the spatial resolution limitations of PPF, this study introduced the SPPF composite method, incorporating fine-scale spatial information from GE imagery through three steps: GE feature extraction, spatial-scale normalization and geometric registration, and image compositing.

First, spectral and textural features were extracted from GE imagery. For spectral features, the Normalized Green-Blue Difference Index (NGBDI) and the Normalized Green-Red Difference Index (NGRDI), derived from the RGB bands (Table

4), have proven effective for wetland vegetation classification (Zheng et al., 2022). Texture features were computed from the red band using the GLCM method (Haralick et al., 2007), extracting four second-order statistics commonly used in vegetation classification: contrast, entropy, correlation, and homogeneity (Wang et al., 2015). Sliding window size is critical to texture extraction, with small windows failing to capture spatial texture and large ones blurring object boundaries. A 17×17 sliding window was applied to the grayscale image of the red band (Li et al., 2020), generating GLCMs and

computing four texture metrics per window to produce corresponding texture bands.

Second, enabling effective integration of multi-source data required resampling to a common resolution and geometric registration. The Sentinel-2 spectral bands and associated vegetation index images (10–20 m) were resampled using cubic



convolution to match the 0.9 m GE imagery. Then, GE imagery was used as the reference to selecting evenly distributed and clearly identifiable control points from both image sources (e.g., tidal creek intersections, aquaculture pond corners, and

vegetation patch boundaries). These points were used to construct a polynomial transformation model for registering the Sentinel-2 imagery. Finally, phenological features derived from Sentinel-2 were integrated with the spectral, texture, and RGB features extracted from GE imagery to construct the SPPF composite images consisting of 24 bands.

**Table 4 The RGB-based spectral indices used in this study.**

| Vegetation index | Formula | Reference |
|---|---|---|
| NGBDI | (Green - Blue) / (Green + Blue) | Du and Noguchi. (2017) |
| NGRDI | (Green - Red) / (Green + Red) | Gitelson et al. (2002) |

**2.3.3 Integration of object-based spatial features in SPPF**

Considering the complex boundaries and homogeneous interiors of *S. alterniflora* patches, accurately delineating their edges remains challenging when using pixel-based features. Therefore, we developed an object-based feature extraction method that incorporated a multi-scale optimized segmentation strategy, enabling the effective integration of spatial context and pixel neighborhood relationships for improved boundary detection. The method included two key steps: identifying the

boundary regions of *S. alterniflora* patches and determining the optimal multi-scale segmentation parameters.

To delineate patch boundaries, each patch identified from the CMSA was expanded outward and contracted inward by 10 m, corresponding to the spatial resolution of the CMSA. This process resulted in a 20 m annular buffer zone that captures the complex transitional areas along the edges of *S. alterniflora* patches while minimizing the inclusion of non-target features, thereby ensuring both the accuracy and representativeness of boundary identification. To implement the multi-scale

optimized segmentation strategy, the Estimation of Scale Parameter (ESP) method was applied to identify the optimal scales for both edge-complex and interior-homogeneous regions. ESP quantifies image region homogeneity by computing Local Variance (LV) and its Rate of Change (ROC) across multiple scales, with ROC peaks typically indicating optimal segmentation (Drăguţ et al., 2010). To improve scale representativeness, one typical region was selected from each of the five subregions (see Sect. 2.1), and GE imagery was segmented to calculate LV and ROC. Following Wang et al. (2021), the

scale range was set to 4-60 with a step of 1. The shape and compactness parameters were set to 0.1 and 0.5, respectively (Wan et al., 2014). As shown in Fig. 4(a), the mean ROC curve across the five regions exhibited multiple peaks, indicating several candidate optimal scales. Based on these peaks, a series of segmentation results were visually interpreted. Scale 16 was optimal for capturing fine details in boundary-complex regions, while scale 24 was better represented the homogeneous interior, balancing spatial coherence with processing efficiency. Segmentation results for both scales are shown in Figs. 4(b)

and 4(c). Based on the determined scale parameters, object-based segmentation was performed on the SPPF composite imagery in eCognition, producing the OSPPF for subsequent classification.





**Figure 4: (a) Variation of Local Variance (LV) and Rate of Change (ROC) with Scale Parameter. (b) Segmentation results with scale parameter of 16. (c) Segmentation results with scale parameter of 24. The VHR imagery in the figure is from © Google Earth 2020.**

### 2.3.4 Classifier selection and parameterization

The object-based RF classifier, which integrates spectral, textural, and spatial contextual features, has been shown to outperform pixel-based RF method in mapping *S. alterniflora* (Tian et al., 2020b; Yan et al., 2021). Therefore, we used the object-based RF classifier in eCognition. The RF algorithm aggregates predictions from multiple decision trees, with the number of trees being a key parameter influencing classification performance (Breiman et al., 2001). To determine the optimal number of trees, a sensitivity analysis was conducted using the training and validation datasets (see Sect. 2.2.3), varying the tree count from 50 to 500 at intervals of 50. The results indicated that 200 trees yielded the highest overall accuracy on the validation set. The multi-source features and training samples were then input into the object-based RF





classifier to produce the initial *S. alterniflora* map. To enhance accuracy, experienced researchers visually interpreted GE
imagery and corrected the initial results. Consequently, the final sub-meter *S. alterniflora* Map of Mainland China (CM-SSM)
was generated.

## 2.4 Accuracy assessment

To assess the mapping effectiveness of the OSPPF method for *S. alterniflora*, two alternative methods were applied in
typical area to generate comparative results. Each result was compared with the CM-SSM generated by OSPPF, focusing on
boundary delineation, small patch detection, and internal structure extraction. Classification accuracy of the CM-SSM was
quantitatively assessed using confusion matrix-based metrics. Producer accuracy (PA), user accuracy (UA), overall accuracy
(OA), and the F1 score were calculated using the validation dataset, which comprised 3,170 positive and 6,075 negative
samples (see Sect. 2.2.3). To further evaluate the performance of CM-SSM, a comparative analysis was conducted against
the CMSA. The comparison included both classification detail and overall statistics. At the detail level, attention was paid to
differences in edge, internal structure, and small patch. At the statistical level, we quantified differences in total area, number
of patches, and spatial distribution of *S. alterniflora*.

## 3 Result

### 3.1 Performance of OSPPF

To assess the contribution of GE imagery to classification performance, *S. alterniflora* mapping was conducted using two
methods in the Dandou Sea: (1) Object-based PPF (OPPF) classification using resampled Sentinel-2 imagery alone, and (2)
OSPPF classification integrating both Sentinel-2 and GE imagery. As shown in Fig. 5(a), classification based solely on
Sentinel-2 imagery was able to capture the general outline of *S. alterniflora* communities but failed to effectively delineate
open spaces within the patches. In addition, Fig. 5(b) demonstrates that small, fragmented *S. alterniflora* patches were poorly
detected, and the boundaries between *S. alterniflora* and mangroves were inaccurately represented. In contrast, the CM-SSM
generated using fused GE imagery exhibited superior spatial detail, successfully identifying small patches and internal
details, as well as accurately delineating boundaries between *S. alterniflora* and co-occurring species.

To further compare pixel-based and object-based classification methods in *S. alterniflora* mapping, the SPPF and OSPPF
methods were applied in the Dandou Sea. As illustrated in Fig. 6, the CM-SSM generated using object-based classification
accurately extracted the boundaries between *S. alterniflora* and surrounding land cover types such as mudflats and
mangroves, whereas the pixel-based method showed poor boundary delineation. Moreover, Fig. 6 indicates that the pixel-
based result suffered from salt-and-pepper noise, particularly within and along the edges of *S. alterniflora* patches. In
contrast, the CM-SSM demonstrated a smoother spatial distribution and effectively suppressed such noise.

**(a)**

**(b)**

GE imagery        OPPF method        OSPPF method

**Figure 5: Comparison of classification results using OPPF and OSPPF methods in Dandou Sea. The VHR imagery in the figure is from ⓒ Google Earth 2020.**




| (a) | | |
|:---:|:---:|:---:|
| (b) | | |
| GE imagery | SPPF method | OSPPF method |

Figure 6: Comparison of classification results using SPPF and OSPPF methods in Dandou Sea. The VHR imagery in the figure is from © Google Earth 2020.

## 3.2 Comparison with the latest 10 m product

### 3.2.1 Accuracy assessment

Table 5 presents the accuracy assessment of CMSA and CM-SSM based on validation samples (see Sect. 2.2.3). CM-SSM achieved the highest classification performance, with an OA of 96.76 % and an F1 score of 0.95, representing improvements of 14.60 % and 0.21 over CMSA, respectively. The significant accuracy gain demonstrates the superior capability of CM-SSM in capturing the spatial distribution of *S. alterniflora*. These results validate the effectiveness of the proposed OSPPF method and highlight the value of generating sub-meter *S. alterniflora* products for precise large-scale mapping.



**Table 5 Classification accuracy assessment results of CMSA and CM-SSM.**

| Product | Class | PA (%) | UA (%) | F1 | OA (%) |
|---|---|---|---|---|---|
| CMSA | *S. alterniflora* | 89.65 | 62.87 | 0.74 | 82.16 |
| | Non-*S. alterniflora* | 79.79 | 92.23 | | |
| CM-SSM | *S. alterniflora* | 98.00 | 92.84 | 0.95 | 96.76 |
| | Non-*S. alterniflora* | 96.16 | 98.80 | | |

### 3.2.2 Spatial details

To reveal the difference between CM-SSM and CMSA, a comparative analysis focusing on small patches, boundaries, and internal structural features of *S. alterniflora* communities was conducted (Fig. 7). First, given the fragmented distribution of
*S. alterniflora*, numerous small patches are typically present. CM-SSM effectively detected nearly all of them, whereas CMSA missed most. Second, *S. alterniflora* often encroaches upon mangrove habitats due to its aggressive spread, increasing boundary complexity. CM-SSM showed superior delineation of the interface between the two species, clearly depicting invasion fronts that CMSA did not capture effectively. Finally, intertidal *S. alterniflora* communities often include internal features like creeks and open spaces. CM-SSM successfully resolved open spaces and narrow creeks, while CMSA
lacked the spatial detail to do so.



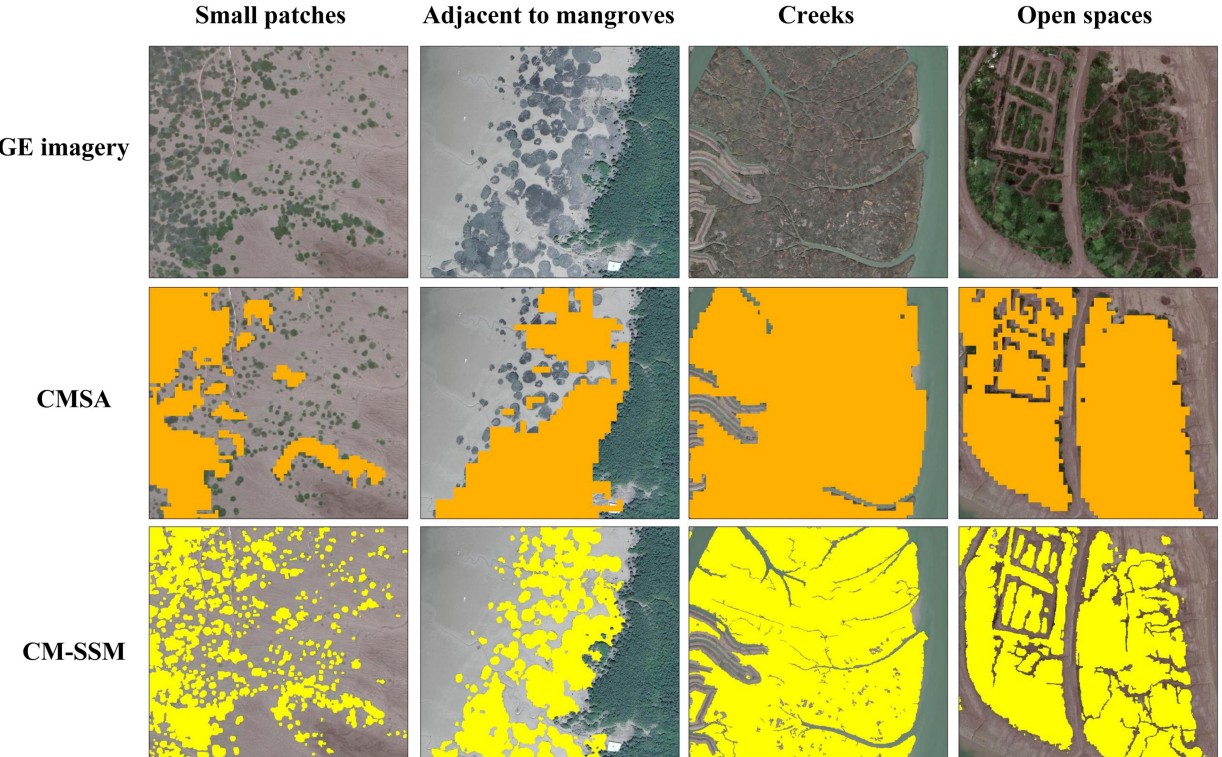

**Figure 7: Spatial detail comparison between CM-SSM and CMSA in typical cases. The VHR imagery in the figure is from ©
Google Earth 2020.**

### 3.2.3 Spatial distribution statistics

A total of 148,072 patches were identified in CM-SSM, approximately 17 times more than CMSA. The total mapped area of
CM-SSM reached 59,371 ha, exceeding that of CMSA by 1,365 ha. To further compare patch size and area differences
between the two products, statistics were summarized across six area classes defined by the minimum mapping unit of
CMSA (i.e., one 10 m pixel). These intervals included: 0.01 ha (1 pixel), 0.1 ha (10 pixels), 1 ha (100 pixels), 100 ha (10,000
pixels), 1,000 ha (100,000 pixels), and greater than 1,000 ha (Fig. 8). In addition, to assess spatial distribution differences, an
overlay analysis was conducted using CM-SSM as the reference (Figs. 9 and 10).

As shown in Fig. 8, when patch area was no greater than 0.1 ha, CM-SSM detected 44 times more patches than CMSA, with
a total area 14 times larger. This notable difference underscores CM-SSM's superior ability to capture small and fragmented
*S. alterniflora* communities. In CM-SSM, such small patches accounted for 90.48 % of the total patch count but only 2.25 %
of the total area. CMSA reported 35.15 % and 0.16 % for the same class. These results are consistent with the highly
fragmented spatial distribution of *S. alterniflora*. In terms of large patches exceeding 100 ha, both products demonstrated
similar performance. In CM-SSM, these patches represented just 0.05 % of the total number but contributed 52.99 % of the
total area, while in CMSA the corresponding proportions were 0.87 % and 62.96 %.

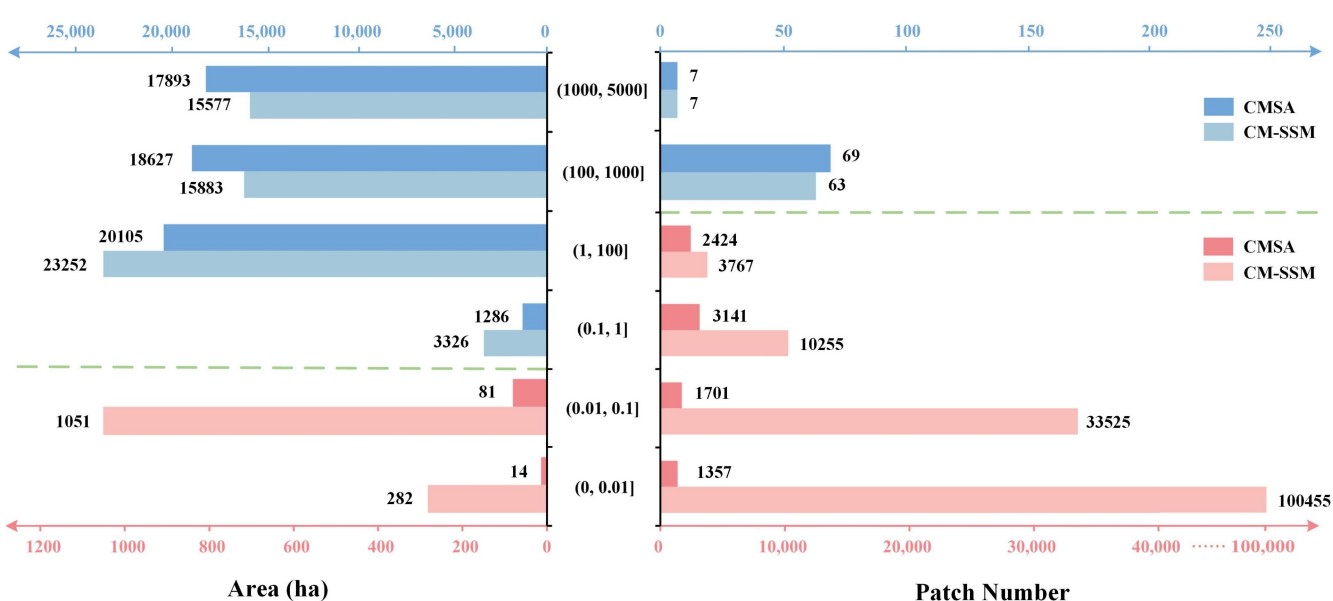

**Figure 8: Zonal statistics of the number and area of *S. alterniflora* patches identified by CM-SSM and CMSA.**

When benchmarked against CM-SSM, CMSA exhibited 16,454 ha of commission and 17,819 ha of omission, resulting in a total spatial mismatch of 34,273 ha (Fig. 9). Notably, the overall area difference between the two products was small because the commission and omission nearly offset each other. To further explore spatial discrepancies, a province-level analysis was conducted. As shown in Fig. 10, both the pattern and magnitude of these differences varied across provinces. For example, in Shanghai, omission was the primary contributor to spatial disagreement, whereas in Jiangsu, commission and omission were

more balanced, together accounting for 7,760 ha difference.

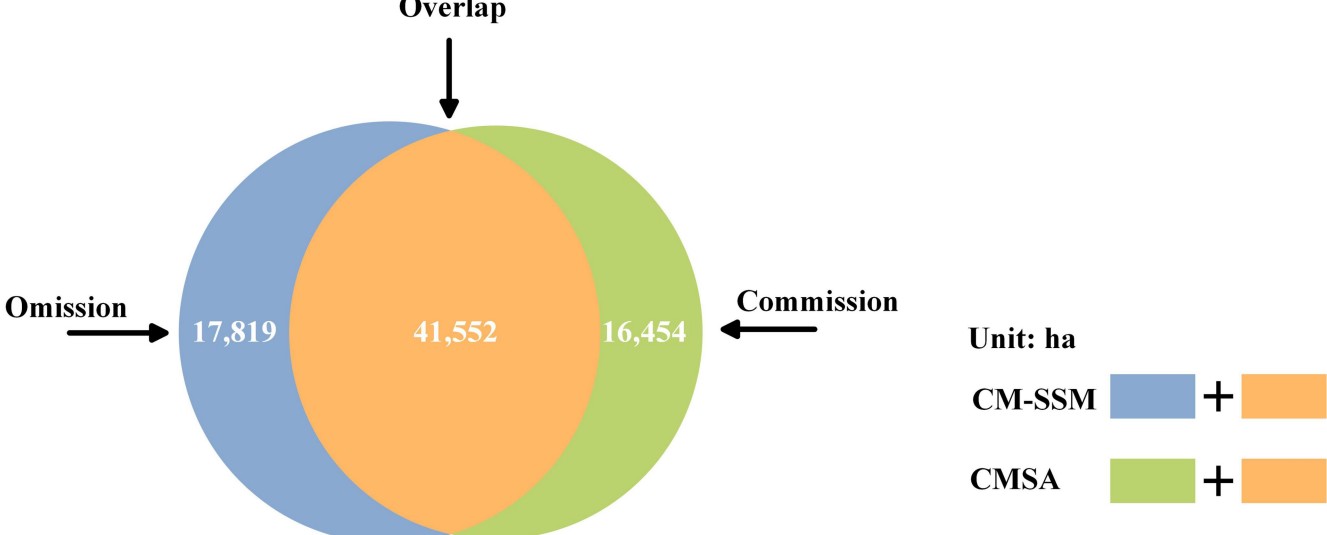

**Figure 9: Spatial distribution difference statistics of *S. alterniflora* identified by CM-SSM and CMSA.**

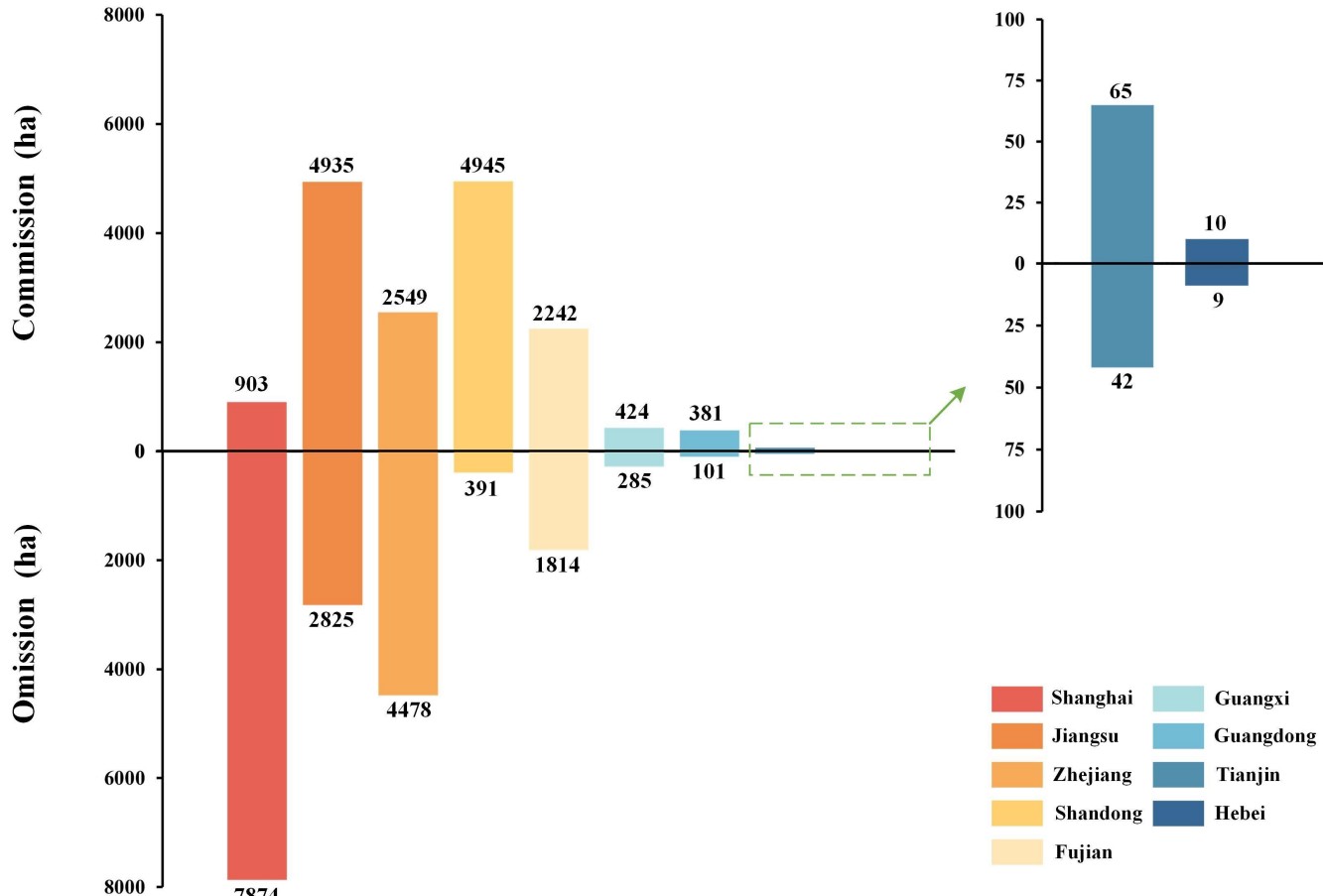

**Figure 10: Using CM-SSM as the reference, the commission and omission of *S. alterniflora* area by province were calculated for CMSA.**

## 3.3 Mapping results of mainland China

Figure 11(a) illustrates the 2020 distribution of *S. alterniflora* across mainland China based on CM-SSM. The results indicate that *S. alterniflora* was densely distributed along the coasts of Fujian, Zhejiang, Shanghai, and Jiangsu, where nearly all large-scale communities were located. These four provinces also reported the largest total areas, together accounting for 94.08 % of the total in mainland China (Fig. 12). In Guangxi, large patches were primarily distributed around the Beibu Gulf, while in Guangdong, *S. alterniflora* appeared more scattered and patchier. In northern provinces such as Shandong, Hebei, and Tianjin, *S. alterniflora* communities were generally small in size and spatially fragmented. To better illustrate spatial patterns across ecosystems, three representative regions were selected. As shown in Figs. 11(b) and 11(d), Dandou Sea and the Zhangjiang Estuary reflect zones of *S. alterniflora*-mangrove coexistence, where *S. alterniflora* is gradually encroaching into mangrove habitats. In contrast, Fig. 11(c) highlights northern Jiangsu's mudflats, where *S. alterniflora* has colonized unvegetated mudflats and formed expansive stands, demonstrating strong invasiveness and ecological adaptability.

Earth System
Discussions
Science
Data

**Figure 11: (a) Spatial distribution of *S. alterniflora* in mainland China for 2020, derived from the CM-SSM, with each grid cell representing the total *S. alterniflora* area within a 10 km × 10 km unit; (b–d) CM-SSM distribution map in typical areas, including 375 (b) Dandou Sea, (c) mudflats of northern Jiangsu, and (d) Zhangjiang Estuary. The background imagery is provided by Esri (https://www.esri.com) and its data partners. The VHR imagery in the figure is from © Google Earth 2020.**





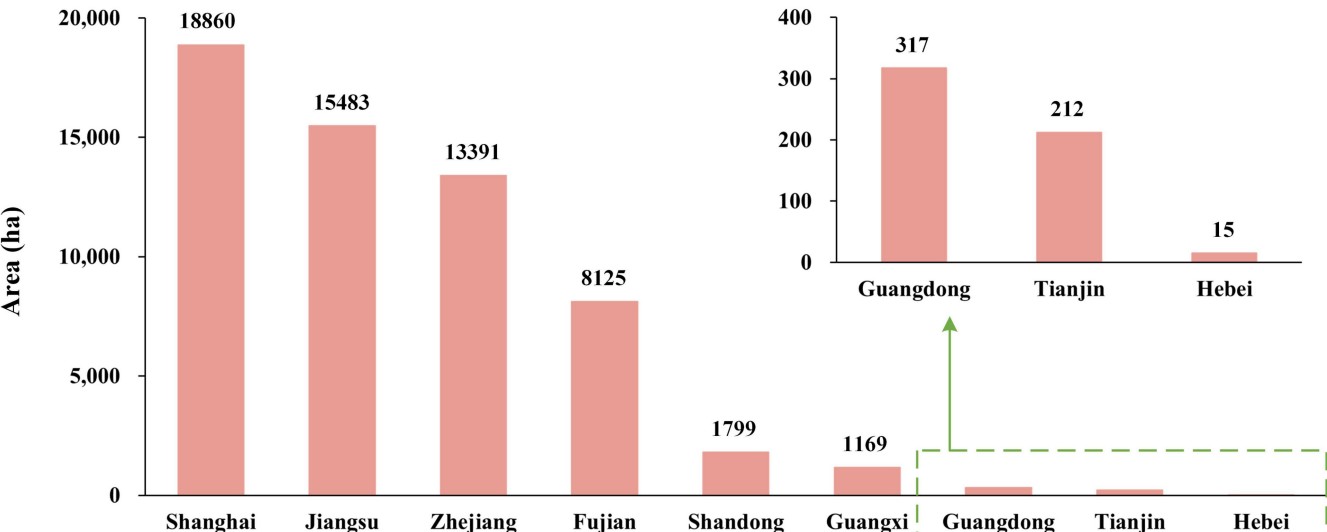

**Figure 12: Provincial statistics of *S. alterniflora* area identified by CM-SSM.**

## 4 Discussion

### 4.1 The advantages of the OSPPF

This study proposed a novel OSPPF composite method that enhanced the PPF by integrating GE imagery and object-based classification. Previous studies have demonstrated that PPF improves the spectral separability of *S. alterniflora* by utilizing dual-phase phenological information (Tian et al., 2020a; Li et al., 2024). However, PPF based solely on Sentinel-2 imagery is limited in capturing spatial detail due to its 10 m resolution, despite providing rich phenological information. To overcome this limitation, GE imagery was integrated to enhance spatial detail (Fig. 5), offering two key advantages. First, GE imagery mitigates the mixed-pixel problem. Patch-level statistics show that 100,455 patches are smaller than 100 m², accounting for 67.84 % of all patches (Fig. 8). These patches are often overlooked in Sentinel-2 classifications due to mixed-pixel problem, yet they are critical indicators of early-stage invasion. In contrast, the sub-meter resolution of GE imagery allows clear visualization of *S. alterniflora* texture and structure, facilitating the accurate delineation of boundaries, internal structure, and small patches. Second, GE imagery provides a more robust foundation for object-based classification. The object-based classification method used in this study relies on OBIA for effective image segmentation, and the quality of segmentation directly affects classification accuracy (Hao et al., 2021). Compared with Sentinel-2, GE imagery more accurately delineates object boundaries and internal structure, enabling the construction of clearly defined and spatially consistent segments that provide a superior basis for classification.

Moreover, the object-based classification applied in this study outperformed the pixel-based method in both accuracy and boundary delineation (Fig. 6). On the one hand, object-based classification reduces uncertainty in identifying *S. alterniflora*.



The GE imagery contains complex spectral and textural information, which increases salt-and-pepper noise and reduces the reliability of pixel-level classification. By aggregating spectrally, texturally, and spatially consistent pixels, object-based classification reduces intra-class noise, improving both robustness and accuracy. On the other hand, the multi-scale

optimized segmentation enhanced boundary delineation. The coexistence of *S. alterniflora* with native vegetation such as mangroves and *Phragmites australis* causes spectral mixing at patch edges, which pixel-based methods struggle to resolve. Although object-based methods improve boundary delineation, segmentation quality remains critical to classification accuracy. Therefore, we proposed a multi-scale optimized segmentation strategy designed to address the spectral heterogeneity at patch boundaries and spectral homogeneity within interiors, aiming to enhance the precision of boundary

extraction. Specifically, a coarse scale was applied to homogeneous interiors, while a finer scale was used for complex boundaries. This strategy improves boundary delineation and avoids over-segmentation in uniform areas, enhancing both classification efficiency and accuracy.

## 4.2 The advantages of CM-SSM

This study produced the first Sub-meter *S. alterniflora* Map for Mainland China in 2020 (CM-SSM). Compared with the

latest 10 m product (CMSA), CM-SSM demonstrated superior performance in classification accuracy, spatial detail representation, and spatial distribution statistics. Specifically, CM-SSM achieved higher classification accuracy, with OA and F1-score improved by 14.60 % and 0.21, respectively (Table 5). These advancements are primarily attributed to the development of sub-meter phenological features for *S. alterniflora* and the adoption of an object-based classification strategy (see Sect. 4.1).

CM-SSM also exhibited significant advantages in spatial detail extraction. Previous studies have indicated that *S. alterniflora* tends to exhibit a fragmented distribution pattern, where numerous small patches are often overlooked due to the limitations of 10 m resolution imagery (Zhou et al., 2024). Additionally, the species is often associated with diverse companion species and interspersed with narrow tidal creeks, which are difficult to distinguish using 10 m resolution due to the mixed pixel problem (Li et al., 2024). By increasing the mapping resolution to 0.9 m, CM-SSM effectively alleviated the

impact of mixed pixels, enabling the accurate identification of small *S. alterniflora* patches, as well as their boundaries and internal structures (Fig. 7). This capability provides critical support for developing invasive species management strategies. For example, areas with dense small patches typically present higher risks of spread, and prioritizing these regions for control measures may help suppress expansion trends (Liu et al., 2016). Furthermore, CM-SSM's ability to delineate growth boundaries between *S. alterniflora* and companion species offers valuable insights into its invasion processes and potential

threats to native species, such as mangroves (Chen and Shi, 2023). In addition, the accurate recognition of tidal creeks within *S. alterniflora* communities provides a data foundation for analyzing the relationship between creek morphology and spatial expansion (Kearney and Fagherazzi, 2016; Sanderson et al., 2000).

In terms of spatial distribution statistics, CM-SSM demonstrated higher accuracy. As shown in Sect. 3.2.3, although CM-SSM identified a significantly greater number of small *S. alterniflora* patches than CMSA, both products exhibited high





consistency in detecting large *S. alterniflora* communities, resulting in a relatively small difference of only 1,365 ha in total area (Fig. 8). However, spatial overlay analysis revealed a substantial spatial discrepancy of up to 34,273 ha between the two products, accounting for 57.73% of the total CM-SSM area (Figs. 9 and 10). This discrepancy is primarily due to the limited spatial resolution of CMSA, which introduces classification errors. On the one hand, mixed pixels result in misclassification of other land types (e.g., creeks and open spaces) as *S. alterniflora*, leading to area overestimation. On the other hand, the

failure to detect highly fragmented small patches leads to area underestimation. These findings underscore the critical role of spatial resolution in accurately capturing the distribution of *S. alterniflora* and highlight the limitations of relying solely on total area statistics, which may obscure substantial differences between products.

Accurate spatial distribution data of *S. alterniflora* are essential for the reliable quantification of its carbon storage (Xia et al., 2021; Xu et al., 2022). In this study, Soil Organic Carbon (SOC) storage was estimated using both CM-SSM and a multi-

temporal 30 m dataset developed by Mao et al. (2019), based on a unified provincial-level SOC unit storage coefficient for *S. alterniflora* (Zhang et al., 2024). The results indicated that the total SOC storage estimated from CM-SSM reached 3,767.69 Gg, which is 706.69 Gg higher than the estimate derived from the 30 m product, with particularly notable differences observed in Shanghai, Zhejiang, and Shandong (Fig. 13). The primary reason for this discrepancy lies in the resolution limitations of the 30 m product, which resulted in omission of small patches and misclassification at the edges, thereby

reducing mapping accuracy. In contrast, CM-SSM provided a more accurate representation of the invasion pattern of *S. alterniflora*, leading to more reliable SOC estimates. This result highlights the value of fine-scale mapping in improving the accuracy of SOC storage estimation.

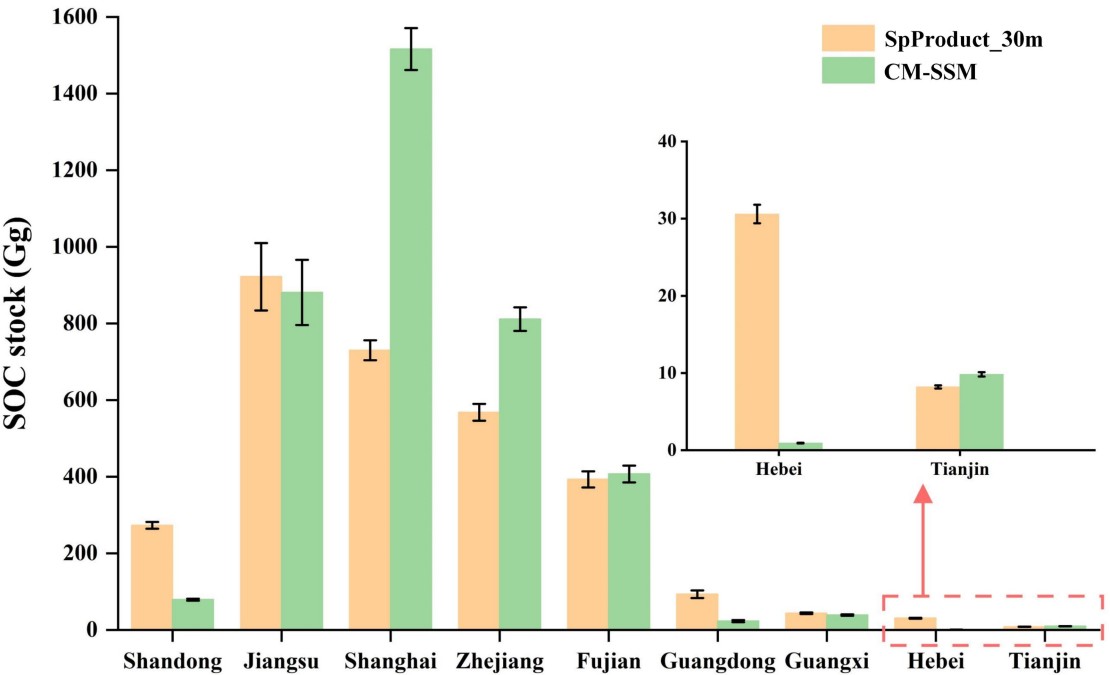

**Figure 13: Estimated SOC stock in the 0–1 m soil layer of *S. alterniflora* across coastal provinces of mainland China in 2020 based on SpProduct_30m and CM-SSM.**

## 4.3 Limitations and prospects

Although this study proposed the OSPPF method and successfully produced the first Sub-meter *S. alterniflora* Map in Mainland China for the year 2020 (CM-SSM), several limitations remain. The OSPPF method has three main limitations. First, the automation of sample points generation requires further improvement. In this study, sample points were generated through visual interpretation based on field surveys and VHR imagery. However, this process is highly dependent on the expertise of researchers and is time-consuming, particularly for large-scale mapping. Although the ASC-CKM method proposed by Tian et al. (2025) leveraged the CascadeKMeans algorithm and the Mangrove Vegetation Index (MVI) for automated sample generation and achieved success in sub-meter mangrove mapping across China, an efficient and widely accepted classification index specific to *S. alterniflora* is currently lacking, which limits the direct transferability of this approach. Future research could focus on developing a highly discriminative spectral index tailored for *S. alterniflora*, with potential to support automated sample generation.

Additionally, the reliance on high-quality GE imagery constrains the broader applicability of the method. The frequent cloud cover of intertidal zones poses significant challenges to acquiring high-quality GE imagery at global scales or over long time series for sub-meter mapping of *S. alterniflora*. To address this issue, super-resolution techniques, which reconstruct high-resolution details from low-resolution imagery, have shown promising potential (Chen et al., 2024).





Third, the object-based classification incorporating multi-scale optimized segmentation still relies on existing large-scale *S. alterniflora* products during implementation. When applied to global-scale or long-term mapping, the reliance on global reference products presents a key limitation. Future research may explore the application of deep learning models to *S. alterniflora* mapping, potentially replacing the object-based approach (Li et al., 2024). It is worth noting that the CM-SSM
product developed in this study has the potential to serve as a valuable benchmark dataset for the training and evaluation of deep learning models.

The CM-SSM product has two limitations. First, temporal inconsistency among imagery sources may lead to classification errors. Some of the GE imagery used in this study was acquired in 2019 and 2021, which introduces temporal discrepancies that can cause misclassification in areas where *S. alterniflora* exhibits significant spatiotemporal dynamics, ultimately
affecting the mapping accuracy. To address this issue, the integration of multi-source remote sensing data could be explored in future studies to mitigate the impact of temporal mismatches and further reduce classification errors. Second, manual visual interpretation introduces subjectivity, which may result in minor inaccuracies. Although this study utilized VHR imagery for visual interpretation to optimize the mapping product, differences in *S. alterniflora* texture, morphology, and spectral appearance across climate zones posed challenges for consistent manual interpretation. Future research could
explore semi-automated methods or interpretation strategies supported by deep learning to reduce uncertainties arising from human intervention.

## 5 Data availability

The 2020 Sub-meter *S. alterniflora* Map of Mainland China (CM-SSM) generated by this study is openly available at https://doi.org/10.5281/zenodo.16296823 (Xu et al., 2025).

## 6 Conclusion

This study proposed a novel Object- and Sub-meter-enhanced Pixel-based Phenological Feature (OSPPF) composite method to generate the first 2020 Sub-meter *S. alterniflora* Map of Mainland China (CM-SSM). The OSPPF method integrates multi-source remote sensing imagery and employs an object-based classification method with a multi-scale optimized segmentation strategy, effectively addressing limitations of existing methods in delineating small patches, boundaries, and
internal details of *S. alterniflora*. Compared with the latest 10 m resolution *S. alterniflora* map (CMSA), the CM-SSM shows significant improvements in classification accuracy, spatial detail, and spatial distribution statistics. Specifically, OA and F1 score of CM-SSM are 14.60 % and 0.21 higher than those of CMSA, respectively. While the total area difference between the two products is only 2.30%, spatial distribution discrepancies reach 57.73 %, and the number of detected patches in CM-SSM is 17 times greater than in CMSA. The CM-SSM product and its underlying OSPPF method provide high-precision
baseline data for monitoring *S. alterniflora*, and offer a scalable framework for future sub-meter mapping at broader spatial



and temporal scales. These advancements hold substantial potential for supporting *S. alterniflora* management effectiveness assessments and blue carbon stock estimations.

## Author contributions

Z.B., X.M., and T.J.: Conceptualization, methodology, formal analysis, data curation, and writing—original draft. J.M., M.D., and C.K.: Conceptualization, investigation, and writing—review and editing. Z.X., J.H., S.J., H.Y., and W.M.: Data curation, validation, and writing—review and editing. K.Y., Z.Z., and Z.L.: Software, visualization, and writing—review and editing. L.X. and G.H.: Supervision, conceptualization, and writing—review and editing.

## Competing interests

The authors declare that they have no conflict of interest.

## Acknowledgments

This work was supported by the National Natural Science Foundation of China (No. 42171330) and the Beijing Natural Science Foundation (No. L251047).

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
