# Peer review of "National-Scale Sub-meter Mapping of *Spartina alterniflora* in Mainland China 2020"

_Earth System Science Data, 2025_

## Author Response (AR1)

Dear Editor and Reviewers:

On behalf of my co-authors, we sincerely appreciate the opportunity to revise our

manuscript entitled "National-Scale Sub-meter Mapping of Spartina alterniflora

in Mainland China 2020" (Manuscript Number: essd-2025-436). We are grateful

for the thoughtful and constructive comments provided, which have significantly

strengthened the scientific rigor, clarity, and overall quality of our work.

In response, we have carefully revised the manuscript, with all changes

highlighted for clarity. Additionally, we provide a point-by-point response to all

reviewer comments, addressing all concerns in detail.

Thank you very much for your thoughtful consideration.

Yours faithfully,

Jinyan Tian, on behalf of all authors

Associate Professor, College of Resource Environment and Tourism

Capital Normal University, Beijing, China

Email: tjyremote@126.com

**Referee#1**

**Overall comment:**

The manuscript presents a significant and novel contribution by producing the first sub-meter resolution map of *Spartina alterniflora* for mainland China. The proposed OSPPF method effectively integrates multi-source data and object-based analysis to address critical limitations of existing products. The study is well-structured, the methodology is sound, and the dataset is highly valuable for the community. However, several major and minor points require clarification and improvement before the manuscript can be considered for publication.

Thank you for your positive evaluation and for the detailed, constructive feedback. We greatly appreciate your insights, which have been invaluable in refining our manuscript. Below, we provide a comprehensive response to each of your comments.

**Main comments:**

1. The use of Google Earth imagery from 2019 and 2021 alongside 2020 Sentinel-2 data introduces a potential source of error. *S. alterniflora* dynamics can be rapid. Please quantify the extent (e.g., percentage of area) where non-2020 imagery was used and discuss the potential impact of this temporal mismatch on classification accuracy and the final area estimate. A sensitivity analysis in these areas would significantly strengthen the manuscript.

**Response:** In the data processing, we prioritized the use of 2020 Sentinel-2 and Google Earth (GE) imagery. For a small number of areas where 2020 imagery was missing or of low quality, GE imagery from 2019 or 2021 were used. When selecting this temporally mismatched imagery, we compared the 2019 and 2021 imagery to not only choose the highest-quality imagery but also assess the distribution of *S. alterniflora* across different years. The results indicate that in the vast majority of areas, the spatial distribution of *S. alterniflora* remained largely unchanged, suggesting that temporal dynamics during this period had minimal impact on the mapping results. Furthermore,

statistical analysis shows that the *S. alterniflora* area in these temporally mismatched regions accounts for only 2.20 % of the total study area; therefore, their influence on classification accuracy and total area estimation is negligible. In Section 4.3 of the revised manuscript, we have made the following revisions: "First, temporal inconsistency among imagery sources may lead to classification errors. Although a comparison with imagery from adjacent years confirmed that the spatiotemporal dynamics of *S. alterniflora* were minor in most areas, and regions with temporal discrepancies accounted for only 2.20 % of the total area, temporal inconsistencies in imagery may still introduce slight errors. To address this issue, the integration of multi-source remote sensing data could be explored in future studies to mitigate the impact of temporal mismatches and further reduce classification errors."

2. The final manual refinement, while understandable for a first-of-its-kind map, introduces subjectivity. Please describe the protocol followed for this manual correction (e.g., number of interpreters, ruleset used, process for resolving disagreements) to ensure consistency. Discussing the potential magnitude of error or bias introduced by this step is crucial for assessing the dataset's reliability.

Response: The manual refinement in this study was carried out by three researchers with extensive field experience and expertise in visual interpretation of remote sensing imagery. To minimize subjectivity, a clear protocol was established: the interpretation was adopted when at least two of the three interpreters agreed; in cases where none of the three were fully confident, a joint discussion was conducted until a consensus was reached. This procedure effectively standardized the decision-making process, reducing individual bias and ensuring reliability of the manual corrections. In addition, the accuracy of the initial classification results (ICM-SSM) was assessed, as detailed in Section 3.2.1 of the manuscript:

"Table 5 presents the accuracy assessment of three products (CMSA, ICM-SSM, and CM-SSM) based on validation samples (see Sect. 2.2.3).

Among them, CM-SSM achieved the best classification performance, with OA and F1 scores of 96.76% and 0.95, respectively. ICM-SSM also performed well, with an OA of 93.36% and an F1 score of 0.90, although slightly lower than those of CM-SSM. This difference is mainly attributed to the manual refinement, which improved boundary delineation and the identification of small patches."

Table 5 Classification accuracy assessment results of CMSA, ICM-SSM and CM-SSM.

| Product | Class               | PA (%) | UA (%) | F1   | OA (%) |
|---------|---------------------|--------|--------|------|--------|
| CMSA    | S. alterniflora     | 80.85  | 62.87  | 0.71 | 82.16  |
|         | Non-S. alterniflora | 82.64  | 92.23  |      |        |
| ICM-SSM | S. alterniflora     | 93.68  | 86.47  | 0.90 | 93.36  |
|         | Non-S. alterniflora | 93.21  | 96.95  |      |        |
| CM-SSM  | S. alterniflora     | 97.58  | 92.84  | 0.95 | 96.76  |
|         | Non-S. alterniflora | 96.36  | 98.80  |      |        |

3. The method relies on existing products (CMSA) for defining the study area and segmentation buffers. This limits its application to regions or time periods where such prior maps are unavailable or inaccurate. Please discuss the transferability of the OSPPF method to other regions without relying on existing *S. alterniflora* products. Could the method be adapted to be more automated and independent?

**Response:** In the revised manuscript, we have further discussed the transferability of applying the OSPPF method to other regions without relying on existing *S. alterniflora* products. We believe that the OSPPF method has potential to be more automated and independent. The relevant discussion has been added to Section 4.3 of the manuscript, as follows:

"Third, the object-based classification incorporating multi-scale optimized

segmentation still relies on existing large-scale *S. alterniflora* products during implementation. When applied to global-scale or long-term mapping, the reliance on global reference products presents a key limitation. To improve the transferability of the OSPPF method in regions lacking prior *S. alterniflora* products, alternative approaches can be adopted, such as generating an initial *S. alterniflora* mask using vegetation indices (e.g., NDVI, EVI), or automatically identifying potential distribution areas based on tidal zones, topography, and other environmental variables."

4. The authors rightly identify DL as a promising future direction and even note that CM-SSM could serve as training data. Given that DL models (e.g., U-Net, Transformers) are now state-of-the-art for many fine-scale land cover mapping tasks, a discussion on why an object-based RF was chosen over a DL approach is necessary. A direct comparison, even on a subset, would greatly strengthen the methodological justification, or the limitations of not using DL should be explicitly acknowledged.

**Response:** In our previous study, we applied improved DeepLabv3+ model to produce a sub-meter mapping of *S. alterniflora* in the Beibu Gulf, Guangxi, China. In that study, we discussed the limitations of applying deep learning at the national scale, which are mainly reflected in the following two aspects:

- 1) **Insufficient high-quality training samples:** The superior performance of deep learning models relies on large volumes of reliable training data for pre-training and model optimization. Prior to generating the sub-meter *S. alterniflora* map for mainland China in this study, obtaining such high-quality samples required experienced researchers to manually interpret imagery, a time-consuming and labor-intensive process that is difficult to scale nationally. In contrast, the object-based RF classifier has lower requirements and acquisition costs for training samples, making it more suitable under the current data conditions.
- 2) Limited model generalization due to image quality differences: Owing to weather conditions and satellite imaging variations, GE imagery exhibits

inconsistencies in spectral and radiometric quality across regions. Consequently, deep learning models trained in one area often suffer from reduced generalization when applied to other regions, leading to decreased cross-regional classification accuracy. The object-based RF classifier used in this study mitigates the impact of image discrepancies by aggregating neighboring similar pixels, demonstrating greater robustness when handling variations in image quality across different regions.

Therefore, in this study we chose the object-based RF classifier to ensure more robust results at the national scale. We have revised Section 2.3.4 of the manuscript to clarify this point as follows:

"Although deep learning has shown potential for sub-meter mapping of *S. alterniflora* at the local scale, its reliance on high-quality training samples and the limited generalization ability of models constrain its application at the national scale (Zhou et al., 2024). In contrast, the object-based RF classifier integrates spectral, textural, and spatial contextual features, demonstrating higher stability and classification accuracy in identifying *S. alterniflora*. Moreover, it outperforms the pixel-based RF method in mapping *S. alterniflora* (Tian et al., 2020b; Yan et al., 2021). Therefore, we used the object-based RF classifier in eCognition."

Reference: Zhou, B., Xu, M., Tian, J., Huang, Y., Song, J., Zhu, L., Zhu, X., Qu, X., Zhang, L., Li, X., and Gong, H.: Mapping the invasive Spartina alterniflora in sub-meter level with improved phenological spectral features and deep learning method, Int. J. Digital Earth, 17, 2434634, doi:10.1080/17538947.2024.2434634, 2024.

5. The reported improvements in OA and F1-score are substantial. However, please support these claims with a statistical test (e.g., McNemar's test) to confirm that the difference in accuracy between CMSA and CM-SSM is statistically significant and not due to chance.

Response: Following your suggestion, we conducted McNemar's test to evaluate whether the accuracy improvement of CM-SSM over CMSA is

statistically significant. The test results indicate a statistically significant difference ( $\chi^2 = 820.22$ , p

Figure 5: The importance of multi-sourced features derived from the RF classifier.

New reference: Zhang, X., Liu, L., Zhao, T., Chen, X., Lin, S., Wang, J., Mi, J., and Liu, W.: GWL\_FCS30: global 30 m wetland map with fine classification system using multi-sourced and time-series remote sensing imagery in 2020,

8. The manuscript mentions masking water pixels (SCL=6) to reduce tidal effects. However, tidal state can significantly influence the appearance and detectability of *S. alterniflora*. Please clarify if the Sentinel-2 compositing process considered tidal height information to select images from a consistent low-tide period, or discuss the potential residual impact of tidal variability on the phenological feature compositing.

**Response:** In the Sentinel-2 compositing process, we carefully considered the influence of tidal variability on the detection of *S. alterniflora*. Specifically, during image selection, we prioritized Sentinel-2 observations acquired under consistent low-tide conditions to exclude the interference of high-tide imagery in subsequent phenological feature compositing. Furthermore, we employed the pixel-based compositing approach proposed by Tian et al. (2020a). In this method, water pixels (SCL = 6) and cloudy pixels were masked to ensure that only valid, high-quality pixels were used within each phenological period. This strategy not only effectively mitigated the adverse effects of cloudy and rainy conditions in intertidal zones but also substantially reduced the residual influence of tidal variability on the phenological composites, ensuring that the resulting imagery best represents low-tide and cloud-free conditions.

Reference: Tian, J., Wang, L., Yin, D., Li, X., Diao, C., Gong, H., Shi, C., Menenti, M., Ge, Y., Nie, S., Ou, Y., Song, X., and Liu, X.: Development of spectral-phenological features for deep learning to understand Spartina alterniflora invasion, Remote Sens. Environ., 242, 111745, doi:10.1016/j.rse.2020.111745, 2020a.

9. The text describing the workflow (Figure 2) could be more precise. Please explicitly state the final number of bands in the PPF, SPPF, and OSPPF feature sets. A clear listing of all input features for the RF model would improve reproducibility.

**Response:** To address this point, we have added clarifications in the manuscript

**as follows:**

1) In Section 2.3.1, we had already described that the PPF feature set was constructed from the original spectral bands and vegetation indices of two phenological periods:

"Based on the two identified phenological periods, the PPF was constructed by integrating vegetation indices and original spectral bands. Specifically, five indices were selected to characterize the phenological periods (Table 3): Normalized Difference Vegetation Index (NDVI), Enhanced Vegetation Index (EVI), Plant Senescence Reflectance Index (PSRI), Normalized Difference Water Index (NDWI), and Land Surface Water Index (LSWI). NDVI, EVI, and NDWI were used during the green period, while PSRI and LSWI characterized the senescence period."

"In addition, five original bands of Sentinel-2 were selected for both phenological periods: B2 (blue), B3 (green), B4 (red), B8 (NIR) and B11 (SWIR 1)."

However, the final number of bands was not explicitly stated in the original text. We have now added the following clarification in Section 2.3.1:

"Finally, the vegetation indices and selected spectral bands for both phenological periods (a total of 15 bands) were integrated to construct the PPF composite images."

2) In Section 2.3.2, we had already specified that the SPPF feature set consists of 24 bands, including the 15 upsampled PPF bands, two spectral indices, four texture features, and the RGB bands extracted from GE imagery. The corresponding description is as follows:

"First, spectral and texture features were extracted from GE imagery. For spectral features, the Normalized Green-Blue Difference Index (NGBDI) and the Normalized Green-Red Difference Index (NGRDI), derived from the RGB bands (Table 4), have proven effective for wetland vegetation classification (Zheng et al., 2022). Texture features were computed from the red band using the GLCM method (Haralick et al., 2007), extracting four

second-order statistics commonly used in vegetation classification: contrast, entropy, correlation, and homogeneity (Wang et al., 2015b)."

"Second, enabling effective integration of multi-source data required resampling to a common resolution and geometric registration. The Sentinel-2 spectral bands and associated vegetation index images (10–20 m) were resampled using cubic convolution to match the 0.9 m GE imagery. Then, GE imagery was used as the reference to selecting evenly distributed and clearly identifiable control points from both image sources (e.g., tidal creek intersections, aquaculture pond corners, and vegetation patch boundaries). These points were used to construct a polynomial transformation model for registering the Sentinel-2 imagery. Finally, phenological features derived from Sentinel-2 were integrated with the spectral, texture, and RGB features extracted from GE imagery to construct the SPPF composite images consisting of 24 bands."

- 3) The OSPPF feature set also consists of 24 bands, but unlike the SPPF, these are object-based features. This was described in the manuscript as follows: "Considering the complex boundaries and homogeneous interiors of *S. alterniflora* patches, accurately delineating their edges remains challenging when using pixel-based features. Therefore, we developed an object-based feature extraction method that incorporated a multi-scale optimized segmentation strategy, enabling the effective integration of spatial context and pixel neighborhood relationships for improved boundary detection." "Based on the determined scale parameters, object-based segmentation was performed on the SPPF composite imagery in eCognition, producing the
- 10. Figures 8, 9, and 10 are critical but lack clarity. The y-axis labels in Fig. 8 are cut off. Fig. 9's Venn diagram is simple but effective; ensure the values are clearly visible.

OSPPF for subsequent classification."

**Response:** In response to comment 7, we added Figure 5 to illustrate the importance of different classification features. Accordingly, Figures 8, 9, and

10 in the original manuscript have been renumbered as Figures 9, 10, and 11 in the revised version.

- 1) Regarding Fig. 9, the y-axis labels appear truncated because the two subplots share the same axis. Specifically, both panels summarize statistics of patch area and patch number within identical intervals. This has been clarified in Section 3.2.3 of the manuscript: "To further compare patch size and area differences between the two products, statistics were summarized across six area classes defined by the minimum mapping unit of CMSA (i.e., one 10 m pixel). These intervals included: 0.01 ha (1 pixel), 0.1 ha (10 pixels), 1 ha (100 pixels), 100 ha (10,000 pixels), 1,000 ha (100,000 pixels), and greater than 1,000 ha (Fig. 9)."
- 2) In addition, Figures 9, 10, and 11 have been replaced with higher-resolution versions, and the values in the Venn diagram of Fig. 10 are now clearly visible. The revised figures are shown below.

Figure 9: Zonal statistics of the number and area of *S. alterniflora* patches identified by CM-SSM and CMSA.

Figure 10: Spatial distribution difference statistics of *S. alterniflora* identified by CM-SSM and CMSA.

Figure 11: Using CM-SSM as the reference, the commission and omission of *S. alterniflora* area by province were calculated for CMSA.

11. The captions for Figures 5, 6, and 7 should explicitly state that the OSPPF result is the final, manually refined CM-SSM product for clarity.

**Response:** In response to comment 7, we added Figure 5 to illustrate the importance of different classification features. Accordingly, Figures 5, 6, and 7 in the original manuscript have been renumbered as Figures 6, 7, and 8 in the

revised version.

- 1) We have revised the captions of Figures 6 and 7 as suggested. The updated captions are as follows:
  - "Figure 6: Comparison of classification results using OPPF and OSPPF methods in Dandou Sea. The result generated by the OSPPF method is the final, manually refined CM-SSM product. The VHR imagery in the figure is from © Google Earth 2020."
  - "Figure 7: Comparison of classification results using SPPF and OSPPF methods in Dandou Sea. The result generated by the OSPPF method is the final, manually refined CM-SSM product. The VHR imagery in the figure is from © Google Earth 2020."
- 2) In addition, Section 2.3.4 of the manuscript already states: "To enhance accuracy, experienced researchers visually interpreted GE imagery and corrected the ICM-SSM. Consequently, the final Sub-meter *S. alterniflora* Map of Mainland China (CM-SSM) was generated." This clearly indicates that CM-SSM is the final product after manual refinement. The original caption of Fig. 8 is: "Figure 8: Spatial detail comparison between CM-SSM and CMSA in typical cases. The VHR imagery in the figure is from © Google Earth 2020." Since Fig. 8 directly compares CM-SSM with CMSA, we believe no additional modification to the caption is necessary.
- 12. The term "sub-meter" is used throughout, but the actual resolution of the final CM-SSM product should be explicitly stated early in the abstract and method (it is 0.9m, as mentioned later). Briefly justify why this specific resolution from GE was chosen.

**Response:** According to your suggestion, we have made the following revisions:

- 1) In the revised abstract, we have explicitly stated that the final CM-SSM product has a spatial resolution of 0.9 m, as follows:
  - "To this end, this study produced the first 2020 national-scale Sub-meter (0.9 m) *S. alterniflora* Map of Mainland China (CM-SSM), using an

object- and sub-meter-enhanced pixel-based phenological feature composite method."

In Section 2.3, we further clarified that the adopted GE imagery has a spatial resolution of 0.9 m, as shown below:

"This study proposed an Object- and Sub-meter-enhanced Pixel-based Phenological Feature (OSPPF) composite method for mapping *S. alterniflora*, including four steps (Fig. 2). First, a Pixel-based Phenological Feature (PPF) was constructed using Sentinel-2 imagery (10 m). Second, spatial and texture features extracted from GE imagery (0.9 m) were integrated to enhance the PPF, resulting in the Sub-meter-enhanced PPF (SPPF). Third, a multi-scale object-based segmentation strategy was used to extract the OSPPF. Finally, a RF classifier was applied to generate the initial result, which was then manually refined to generate the final *S. alterniflora* distribution map."

2) Our choice of the 0.9 m resolution was based on two main considerations. First, the 0.9 m resolution is sufficient to effectively capture the small patches and boundary details of *S. alterniflora*, meeting the sub-meter spatial detail required for this study. Although using imagery with even higher resolution could provide finer spatial details, it would substantially increase data redundancy as well as computational and storage costs. Second, our previous study has produced a national-scale mangrove mapping product at a spatial resolution of 0.9 m (Tian et al., 2025). To ensure spatial consistency in the construction of a coastal wetland vegetation dataset and to facilitate subsequent research based on this dataset, the same resolution of 0.9 m was adopted in this study for the mapping of *S. alterniflora*.

Reference: Tian, J., Wang, L., Diao, C., Zhang, Y., Jia, M., Zhu, L., Xu, M., Li, X., and Gong, H.: National scale sub-meter mangrove mapping using an

augmented border training sample method, ISPRS J. Photogramm. Remote Sens., 220, 156 - 171, doi:10.1016/j.isprsjprs.2024.12.009, 2025.

**Referee#2**

**Overall comment:**

The manuscript presents an OSPPF method that effectively integrates multi-source data to generate a sub-meter resolution distribution map of *Spartina alterniflora*. The work is thorough, the methodology is reliable and rigorous, and the resulting dataset holds significant value for the scientific community. However, several issues require clarification or improvement:

Thank you for your positive evaluation and for the detailed, constructive feedback. We greatly appreciate your insights, which have been invaluable in refining our manuscript. Below, we provide a comprehensive response to each of your comments.

**Main comments:**

1. The authors determine phenological transitions based on NDVI and propose thresholds of 0.3 and 0.5. However, in Figure 3, the Y-axis lacks tick marks corresponding to these threshold values. The authors are advised to add tick marks or include horizontal reference lines to improve readability.

**Response:** In Figure 3, we have added horizontal reference lines at NDVI values of 0.3 and 0.5 to indicate the thresholds. The revised Figure 3 is shown below.

Figure 3: NDVI time series analysis of the *S. alterniflora* in JSCZ. Point density is represented using hexagonal binning, with color intensity indicating the concentration of data points.

2. How were the thresholds of NDVI < 0.3 for the senescence period and NDVI > 0.5 for the green period determined? Were they derived from statistical distributions, field observations, or a specific phenological model? Clarification on the rationale behind these thresholds is needed.

**Response:** Tian et al. (2020a) constructed an annual NDVI time series for *S. alterniflora* in the Beibu Gulf and identified two key phenological periods. Before day 132 of the year, NDVI values were consistently below 0.3, representing the senescence period, while during days 164–260, NDVI values were higher than 0.4, corresponding to the green period. Similarly, this study constructed an NDVI-based phenological curve for *S. alterniflora* along the Jiangsu Coastal Zone (JSCZ), showing a consistent temporal pattern with that reported by Tian et al. (2020a) (Fig. 3). To determine NDVI thresholds suitable for this study area, we plotted the NDVI frequency distribution histogram and its first derivative curve for *S. alterniflora* in the JSCZ. Based on the statistical

distribution characteristics and previous research experience, thresholds of 0.3 and 0.5 were selected (Fig. S1). The rationale is as follows:

- 1) Senescence period threshold (NDVI < 0.3): In the NDVI range of 0.28–0.3, the first derivative declines rapidly to its lowest value, indicating a transition in phenological characteristics. If the threshold were set higher than 0.3, transitional pixels would be included, introducing analytical uncertainty. Thus, NDVI < 0.3 was defined as the senescence period threshold to ensure phenological representativeness of the selected pixels.
- 2) Green period threshold (NDVI > 0.5): In the NDVI range of 0.5–0.52, the first derivative increases sharply, indicating a rapid rise in the number of pixels with NDVI values above 0.5. These pixels correspond to dense and healthy *S. alterniflora* stands. Therefore, NDVI > 0.5 was defined as the green period threshold to represent the vigorous growth stage.

We have added an explanation of the basis for threshold selection in Section 2.3.1 of the revised manuscript, as shown below:

"As shown in Fig. 3, the NDVI time series of JSCZ exhibited a phenological pattern consistent with that reported by Tian et al. (2020a). To determine the two key phenological periods of *S. alterniflora* in the JSCZ, the annual NDVI frequency distribution histogram and its first derivative curve were generated based on 175 pure *S. alterniflora* pixels (Fig. S1 in the Supplement). As shown in Fig. S1, the NDVI values exhibited a marked decline around 0.3 and a sharp increase around 0.5, corresponding to the transitions from the senescence to the transitional period and from the transitional to the green period, respectively. Therefore, NDVI values below 0.3 during DoY 1–125 indicated the senescence period, whereas values above 0.5 during DoY 190–325 corresponded to the green period."

Figure 3: NDVI time series analysis of the *S. alterniflora* in JSCZ. Point density is represented using hexagonal binning, with color intensity indicating the concentration of data points.

Figure S1: NDVI frequency distribution and first derivative curve for determining key phenological thresholds of *S. alterniflora* along the Jiangsu Coastal Zone.

Reference: Tian, J., Wang, L., Yin, D., Li, X., Diao, C., Gong, H., Shi, C., Menenti, M., Ge, Y., Nie, S., Ou, Y., Song, X., and Liu, X.: Development of spectral-phenological features for deep learning to understand Spartina alterniflora invasion, Remote Sens. Environ., 242, 111745, doi:10.1016/j.rse.2020.111745, 2020a.

3. Table 5 shows that CM-SSM achieves significant improvements over CMSA in terms of F1 score and overall accuracy (OA). However, the current evaluation of classification accuracy relies on a limited set of metrics. It is recommended that the authors construct confusion matrices to compare the composition of error types (e.g., omission and commission errors) between the two methods, thereby providing deeper insight into the specific aspects in which CM-SSM outperforms CMSA.

**Response:** Following your suggestion, we have added the confusion matrices of CM-SSM and CMSA in Section 3.2.1 (see Table 6) and provided further explanations. The newly added content in the manuscript is as follows:

"The superior performance of CM-SSM is mainly due to its reduction in both omission and commission errors, which contributed to the higher OA and F1 scores, as shown in Table 6."

Table 6 Confusion matrices of CM-SSM and CMSA based on validation samples.

| Product | Reference class     | Pred            | Total               |       |
|---------|---------------------|-----------------|---------------------|-------|
|         |                     | S. alterniflora | Non-S. alterniflora | Total |
| CMCA    | S. alterniflora     | 1993            | 1177                | 3170  |
| CMSA    | Non-S. alterniflora | 472             | 5603                | 6075  |
| CM-SSM  | S. alterniflora     | 2943            | 227                 | 3170  |
|         | Non-S. alterniflora | 73              | 6002                | 6075  |

4. Given that the study area spans a considerable latitudinal range, there may be substantial heterogeneity in phenological characteristics across regions.

Consequently, spatial variability in classification performance should be considered. Currently, the evaluation appears to rely solely on CM-SSM as a reference for calculating omission and commission errors for CMSA. The authors are encouraged to establish a validation dataset spanning multiple latitudinal zones and use it to comparatively assess the performance of CM-SSM and CMSA, demonstrating whether CM-SSM exhibits generalizability across diverse geographical regions.

**Response:** We conducted an additional accuracy assessment to examine the spatial variability of classification performance across different latitudinal regions. Specifically, we evaluated the accuracy of CMSA and CM-SSM using validation samples from five subregions. The corresponding results are presented in the Supplementary Material (Tables S1–S5), and the revised manuscript (Section 3.2.1) has been updated accordingly. The updated content is as follows:

"Given the wide latitudinal span of the study area, the spatial variability in classification performance was further examined by evaluating CMSA and CM-SSM across five subregions using validation samples. The results show that CM-SSM consistently achieved superior performance, with OA exceeding 95.00% and F1 scores above 0.90 in all subregions (Tables S1–S5 in the Supplement)."

Table S1 Classification accuracy assessment results of CMSA and CM-SSM in the Northern Coastal Zone.

| Product | Class               | PA (%) | UA (%) | F1   | OA (%) |
|---------|---------------------|--------|--------|------|--------|
| CMSA    | S. alterniflora     | 84.00  | 72.41  | 0.78 | 87.63  |
|         | Non-S. alterniflora | 88.89  | 94.12  |      |        |
| CM-SSM  | S. alterniflora     | 96.30  | 89.66  | 0.93 | 95.88  |
|         | Non-S. alterniflora | 95.71  | 98.53  |      |        |

Table S2 Classification accuracy assessment results of CMSA and CM-SSM in the Yellow River Delta Coastal Zone.

| Product | Class               | PA (%) | UA (%) | F1   | OA (%) |
|---------|---------------------|--------|--------|------|--------|
| CMSA    | S. alterniflora     | 86.43  | 72.89  | 0.79 | 86.83  |
|         | Non-S. alterniflora | 86.99  | 94.06  |      |        |
| CM-SSM  | S. alterniflora     | 98.11  | 93.98  | 0.96 | 97.33  |
|         | Non-S. alterniflora | 96.94  | 99.06  |      |        |

Table S3 Classification accuracy assessment results of CMSA and CM-SSM in the Jiangsu Coastal Zone.

| Product | Class               | PA (%) | UA (%) | F1   | OA (%) |
|---------|---------------------|--------|--------|------|--------|
| CMSA    | S. alterniflora     | 64.67  | 59.33  | 0.62 | 77.45  |
|         | Non-S. alterniflora | 82.50  | 85.54  |      |        |
| CM-SSM  | S. alterniflora     | 93.77  | 92.05  | 0.93 | 95.66  |
|         | Non-S. alterniflora | 96.48  | 97.27  |      |        |

Table S4 Classification accuracy assessment results of CMSA and CM-SSM in the Yangtze River Estuary Coastal Zone.

| Product | Class               | PA (%) | UA (%) | F1   | OA (%) |
|---------|---------------------|--------|--------|------|--------|
| CMSA    | S. alterniflora     | 83.33  | 8.40   | 0.15 | 72.11  |
|         | Non-S. alterniflora | 71.76  | 99.28  |      |        |
| CM-SSM  | S. alterniflora     | 99.02  | 84.87  | 0.91 | 95.23  |
|         | Non-S. alterniflora | 93.92  | 99.64  |      |        |

Table S5 Classification accuracy assessment results of CMSA and CM-SSM in the Southern Coastal Zone.

| Product | Class               | PA (%) | UA (%) | F1   | OA (%) |
|---------|---------------------|--------|--------|------|--------|
| CMSA    | S. alterniflora     | 80.47  | 55.56  | 0.66 | 79.66  |
|         | Non-S. alterniflora | 79.41  | 92.71  |      |        |
| CM-SSM  | S. alterniflora     | 97.99  | 92.76  | 0.95 | 96.79  |
|         | Non-S. alterniflora | 96.20  | 98.97  |      |        |

5. While the authors employ a variety of features for classification, they do not discuss the relative contribution of each feature to classification performance. It is recommended to analyze and report feature importance using the built-in measures from the Random Forest classifier. Such an analysis would enhance understanding of the key factors driving *Spartina alterniflora* identification and provide valuable insights for future method development.

**Response:** We have added a feature importance analysis using the Random Forest classifier, which is described in Section 3.1 of the revised manuscript. The updated content is as follows:

"To assess the contribution of GE imagery to classification performance, S. alterniflora mapping was conducted using two methods in the Dandou Sea: (1) Object-based PPF (OPPF) classification using resampled Sentinel-2 imagery alone, and (2) OSPPF classification integrating both Sentinel-2 and GE imagery. As shown in Fig. 6(a), classification based solely on Sentinel-2 imagery was able to capture the general outline of S. alterniflora communities but failed to effectively delineate open spaces within the patches. In addition, Fig. 6(b) demonstrates that small, fragmented S. alterniflora patches were poorly detected, and the boundaries between S. alterniflora and mangroves were inaccurately represented. In contrast, the CM-SSM generated using fused GE imagery exhibited superior spatial detail, successfully identifying small patches and internal details, as well as accurately delineating boundaries between S.

alterniflora and co-occurring species. Furthermore, we conducted a feature importance analysis using the RF classifier (Zhang et al., 2022b). As shown in Fig. 5, spectral and texture features derived from GE imagery consistently contributed highly to the classification of *S. alterniflora*. This can be primarily attributed to the rich spatial texture information provided by GE imagery, which effectively complements the phenological features and thereby enhances classification accuracy."

Figure 5: The importance of multi-sourced features derived from the RF classifier.